# Prediction of Human Inhibition Brain Function with Inter-Subject and Intra-Subject Variability

**DOI:** 10.3390/brainsci10100726

**Published:** 2020-10-13

**Authors:** Rupesh Kumar Chikara, Li-Wei Ko

**Affiliations:** 1Department of Biological Science and Technology, College of Biological Science and Technology, National Chiao Tung University, Hsinchu 300, Taiwan; rupesh.bt01g@g2.nctu.edu.tw; 2Center for Intelligent Drug Systems and Smart Bio-Devices (IDS2B), National Chiao Tung University, Hsinchu 300, Taiwan; 3Institute of Bioinformatics and Systems Biology, National Chiao Tung University, Hsinchu 300, Taiwan; 4Drug Development and Value Creation Research Center, Kaohsiung Medical University, Kaohsiung 807, Taiwan

**Keywords:** response inhibition, electroencephalography, machine learning, classification, prediction, frontal cortex, inter-subject variability, intra-subject variability

## Abstract

The stop signal task has been used to quantify the human inhibitory control. The inter-subject and intra-subject variability was investigated under the inhibition of human response with a realistic environmental scenario. In present study, we used a battleground scenario where a sniper-scope picture was the background, a target picture was a go signal, and a nontarget picture was a stop signal. The task instructions were to respond on the target image and inhibit the response if a nontarget image appeared. This scenario produced a threatening situation and endorsed the evaluation of how subject’s response inhibition manifests in a real situation. In this study, 32 channels of electroencephalography (EEG) signals were collected from 20 participants during successful stop (response inhibition) and failed stop (response) trials. These EEG signals were used to predict two possible outcomes: successful stop or failed stop. The inter-subject variability (between-subjects) and intra-subject variability (within-subjects) affect the performance of participants in the classification system. The EEG signals of successful stop versus failed stop trials were classified using quadratic discriminant analysis (QDA) and linear discriminant analysis (LDA) (i.e., parametric) and K-nearest neighbor classifier (KNNC) and Parzen density-based (PARZEN) (i.e., nonparametric) under inter- and intra-subject variability. The EEG activities were found to increase during response inhibition in the frontal cortex (F3 and F4), presupplementary motor area (C3 and C4), parietal lobe (P3 and P4), and occipital (O1 and O2) lobe. Therefore, power spectral density (PSD) of EEG signals (1-50Hz) in F3, F4, C3, C4, P3, P4, O1, and O2 electrodes were measured in successful stop and failed stop trials. The PSD of the EEG signals was used as the feature input for the classifiers. Our proposed method shows an intra-subject classification accuracy of 97.61% for subject 15 with QDA classifier in C3 (left motor cortex) and an overall inter-subject classification accuracy of 71.66% ± 9.81% with the KNNC classifier in F3 (left frontal lobe). These results display how inter-subject and intra-subject variability affects the performance of the classification system. These findings can be used effectively to improve the psychopathology of attention deficit hyperactivity disorder (ADHD), obsessive-compulsive disorder (OCD), schizophrenia, and suicidality.

## 1. Introduction

Over the previous years, researchers have shown increased interest in human response inhibition function. Response inhibition has been measured to be a key component of executive control and decision-making [1,2]. Response inhibition is a process of executive control. The inhibition model refers to the suppression of inappropriate actions, which supports flexible and goal-directed behavior in real environments [3,4]. The human response inhibition (inhibitory control) has been ingrained in our daily life activities including riding a scooter, driving a car, walking, shooting, and games. In real-world, there are many examples of the human inhibition, such as stop a scooter during the red traffic signal when a car comes around the corner without noticing. The stop signal task (SST) and go/no-go (GNG) tasks are popular tool for the study of inhibition in laboratory-scale [3,4]. In this study, we used the stop-signal task to classify the neural activities of human inhibition using inter- and intra-subject variability. Previous studies reported that the stop-signal task is the most appropriate model to investigate the human response inhibition [5,6]. In stop signal task, each participant responded to a visual stimulus, after presenting random trials of visual stimulus, a stop-signal stimulus was presented indicating that the response should be inhibited. The stop signal task has been widely adopted as a way to parametrically quantify the response inhibition.

Former studies used the simple symbol as go and stop signals to explore the neural activities of human response inhibition [7,8,9]. Previous studies did not use the realistic environmental scenario for the participants when they performed the stop-signal task [7,8,9]. Through simple symbol experiment design, we can understand the neural mechanisms of response inhibition. However, in real environment, the situations for human action and inhibition are more complex than performing experiment in laboratory scale. How to classify neural activities of response inhibition in real world situations is more complex. Therefore, in this study, we used battleground scenario to classify EEG signals of human inhibitory control in realistic environmental settings. Where a sniper-scope view was the background, a target picture was the go signal, a nontarget picture was the stop signal. The battleground scenario created a threatening environment and allowed the evaluation of how participants’ response inhibition manifest in this realistic stop-signal task. The electroencephalography (EEG) recordings were acquired to observe the brain activities of inhibitory control with the high temporal resolution [10].

Former neuroimaging studies displayed that the neural activities of inter- and intra-subject variability can be quite different [11,12]. For example, the occipital lobe shows relatively high intra-subject brain connectivity variability but relatively low inter-subject brain connectivity variability [11]. Consequently, investigated brain connectivity variability in the occipital lobe might be wrongly recognized to inter-subject variability of brain connectivity, rather than just intra-subject variability. Most previous EEG studies only reported between-subjects analysis (inter-subject) [7,8,9]. Therefore, in this study, we explore the classification model of inter-subject and intra-subject variability under human inhibitory control. This model can be utilized to predict the inhibition-related mental disorders, such as attention deficit hyperactivity disorder (ADHD), obsessive-compulsive disorder (OCD), schizophrenia, and suicidality.

The EEG neural activities of human inhibitory control have been investigated in right inferior frontal cortex (IFC) and the presupplementary motor area (pre-SMA) [9,13]. The delta (1–4 Hz) and theta (4–7 Hz) band powers have been observed to be increased in the frontal cortex (200–600 ms) under response inhibition [8]. Moreover, the event-related potential (ERP) of N200 and P300 waves were investigated during human inhibitory control in the frontal cortex [8]. Accordingly, previous neuroimaging study found that the right frontal cortex and presupplementary motor area (pre-SMA) have been involved in human inhibitory control [13,14,15]. The parietal lobe plays a functional role in the integration of sensory information from various regions of the human brain [16]. In addition, the parietal lobe has been related to the perception of emotions in facial recognition. Moreover, the parietal lobe is related to the visual stimuli that is less associated to response inhibition [14]. The parietal lobe receives somatosensory and visual information through motor signals and controls the movement of the hand [17]. Furthermore, occipital lobe was associated to the visual perception that is less specifically connected to response inhibition [14,15]. In this study, human inhibition-related EEG neural markers ERP-N200 and ERP-P300 waves were observed using ERP analysis in the F3, F4, C3, C4, P3, P4, O1, and O2 channels. Therefore, these eight EEG channels including F3, F4, C3, C4, P3, P4, O1, and O2 were used as regions of interest to classify the EEG signals of successful stop versus failed stop trials.

In this study, we used parametric machine learning algorithms including the quadratic discriminant analysis (QDA) and the linear discriminant analysis (LDA) classifiers to classify the EEG signals of successful stop versus failed stop trials. In addition, the nonparametric machine learning algorithms were used including the K-nearest neighbor classifier (KNNC) and Parzen density-based classifier (PARZEN) to classify the EEG signals of successful stop versus failed stop trials. We utilized simple nonparametric and parametric machine learning algorithms to estimate model performance; these algorithms are very easy to use. If these simple machine learning algorithms achieve better performance with high accuracy, then the use of more complex classifiers algorithms can be avoided.

The first goal of this study is to explore the neural marker of human response inhibition under the battleground scenario. The second goal of this study is to design a classification model for successful stop versus failed stop trials with inter-subject and intra-subject variability. We hypothesized that both inter-subject and intra-subject variability affect the performance of the classification model when performing brain–computer interface (BCI) technology. Finally, we develop a classification model to predict the successful stop versus failed trails EEG signals using the power spectral density as input features with machine learning nonparametric (KNNC and PARZEN) and parametric (LDA and QDA) classifiers. This study shows that how inter-subject and intra-subject variability affect the performance of the classification system. This model can be used to predict the inhibition-related biomarkers for mental disorders patients, like ADHD, OCD, and schizophrenia. In addition, it can be utilized in BCI technology.

## 2. Materials and Methods

### 2.1. Participants

Twenty subjects (14 men and 6 women; mean age = 23.00; SD = 0.9) participated in the battlefield scenario. All healthy subjects were right-handed and had normal vision. They had no experience in battlefield scenario or stop signal task. None of the subjects had a history of gastrointestinal, cardiovascular, and neurological disorders. Each subject provided written informed consent before participating in the experiment. This study was carried out in accordance with the recommendations of the Institutional Review Board (IRB) of the National Taiwan University, Taipei, Taiwan. The study was approved by the Research Ethics Committee of the National Taiwan University, Taipei, Taiwan, NTU-REC No: 201210HS007.

### 2.2. Experimental Design

In this study, we used a realistic experimental scenario to investigate the human response inhibition. The designed battleground scenario or translational scenario was a modified stop-signal task [6,18], in which the fixation sign, go cue, and stop stimuli were replaced with images of a sniper scope, target, and nontarget. The designed translational scenario has a virtual battleground for the participants, as shown in Figure 1. In this scenario, each subject played the role of the soldier to open fire on target and to hold the fire when the nontarget stimuli appeared. Each subject was instructed to respond or withhold their action by key press of left mouse within 1 s in go trials and stop trials. The EEG signals were collected during go and stop trials. The stop signal delay (SSD), which was around 50% probability of a successful stop (SS), was measured using a staircase tracking system before they executed formal experimental trials. The staircase tracking system operated in the following way: the SSD started at 150 ms and if the participants successfully withhold their response, the SSD would be increased by 50 ms. In failed stop, SSD would be reduced 50 ms and the lower bound of SSD was fixed 150 ms. The basic response inhibition-related parameters like reaction time (RT) in go trial and stop signal reaction time (SSRT) in stop trial were investigated according to the former study of human inhibition [19]. In the battleground scenario, each subject had 200 trials of which 25% were stop trials, whereas the rest 75% were go trials. The total number of trials acquired were 20 × 200 = 4000 in go trials and stop trials. The battleground scenario might provide participants with stronger motivation and real-world experience in the inhibition study based on a virtual scenario. The battlefield scenario gave the participants greater motivation and real-world experience of inhibiting the human response.

### 2.3. Acquisition of Electroencephalography (EEG) Signals

The EEG signals were recorded using a Neuro Scan NuAmps Express system (Compumedics USA Inc., Charlotte, NC, USA) with 32-channels EEG cap, as shown in Figure 2. All EEG signals were examined using EEGLAB software (10.2.2.4b Version, UC San Diego, Swartz Center for Computational Neuroscience (SCCN), La Jolla, CA, USA) and MATLAB R2012b (The MathWorks Inc., Natick, MA, USA) [10]. In EEG cap, all 32-channels were positioned according to the international 10–20 system. The EEG signals were acquired at a sampling frequency rate of 500 Hz. We use an infinite impulse response (IIR) filter to eliminate linear trends in EEG signals. The EEG signals were filtered with a 1–50 Hz band pass IIR filter. The filter configuration was set to 1 Hz high pass and 50 Hz low pass to eliminate high frequency noise. The EEG signals that were considerably contaminated by artifacts, such as muscle activity, eye blinking, eye movement, and environmental noise were first removed manually and then by independent component analysis (ICA) to minimize their influence on the analysis of the EEG signal. In the artifact removal analysis, we found that about 12% of the epochs (trials) were so noisy in the raw EEG signals. Consequently, 12% noisy epochs were rejected from the raw EEG signal. We removed various artifacts, like muscle activities and eye blinking [10,20,21].

### 2.4. Independent Component Analysis (ICA) and Back-Projection

Independent component analysis (ICA) is an effective technique for removing several types’ artifacts, e.g., eye movement, eye blinking, and muscle artifacts [22,23]. Thus, ICA is a useful technique for extracting the clean EEG signals [10,20,21]. To identify numerous types of artifacts in EEG signals, we checked the scalp map, power spectrum of each independent component, as shown in Figure 3. Based on these standards, we separated clean and noisy independent component, back-projecting the retained independent components to clean the EEG signal (i.e., EEG channels), as displayed in Figure 3. In addition, the theory of independent component analysis (ICA) and back-projecting the retained independent components to EEG channels have been used according to the previous study of EEG signal analysis [20,21]. Afterward, we utilized the clean EEG signals to perform the EEG channel-based event-related potential (ERP), event-related spectral perturbation (ERSP), and power spectral density (PSD) analysis, using functions of the MATLAB R2012b and EEGLAB toolbox [10].

### 2.5. Analysis and Epoch Extraction of EEG Signals

Each epoch was extracted from −500 to 1300 ms. The acquired clean EEG signals were divided according to event markers in successful go, successful stop, and failed stop epochs from −500 to 1300 ms. The extracted epoch EEG signals from −500 to 1300 ms were used to measure the event-related potential (ERP) and event-related spectral perturbation (ERSP) analysis. The ERP and ERSP were used as a biomarker for inhibition of human response. The distribution of go trials was 75% and stop trial was 25% across subjects. After removing the artifact epochs (trials), the total number of 3538 trials was observed for ERP, ERSP, and machine learning analysis. To produce the ERP, ERSP, and machine learning results, we used the successful go trials (2522), successful-stop trails (424), and failed-stop trails (592). According to previous ERP and ERSP inhibition studies, we found strong changes in EEG activities after stimulus onset from 1 to 500 ms during go, successful stop, and failed stop conditions [10,14,15]. Therefore, EEG signals were segmented after the onset of the stimulus from 1 to 500 ms to measure power spectral density (PSD) of EEG signals with inter-subject (between-subjects) and intra-subject (within-subjects) variability. We used PSD as an input feature for the classification model under inter- and intra-subject variability. Figure 4 shows all steps of EEG signal analysis and classification of successful stop versus failed stop.

### 2.6. Power Spectral Density (PSD) Analysis

Each epoch was separately transformed and normalized into the time frequency domain using the event-related spectral perturbation (ERSP) routine [10]. A power spectral density (PSD) is the measure of EEG signal’s power content versus frequency. We measured the PSD of EEG signals using periodogram power spectral density estimate functions in the MATLAB R2012b [10]. The EEG signals (1–50 Hz) were used from 1 to 500 ms to observe the power spectrum density under successful stop and failed stop trials. The PSD of all subjects was measured at the frontal lobe (F3 and F4), the motor cortex (C3 and C4), the parietal lobe (P3 and P4), and the occipital lobe (O1 and O2) of the brain using EEG signals of successful stop and failed stop trials. The asterisk shows significant difference between successful stop trials and failed stop trials using the Wilcoxon signed-rank test (*p* < 0.05) at inter-subject and intra-subject variability.

### 2.7. Machine Learning Classifiers: Nonparametric (KNNC and PARZEN) and Parametric (LDA and QDA)

Most of the previous EEG studies used either nonparametric or parametric machine learning algorithms to evaluate the participant’s performance [24,25,26]. However, in our study, for the first time, we compared both nonparametric and parametric machine learning algorithms using EEG signals of successful stop and failed stop trials. In this study, four types of classifiers were utilized to compare the classification accuracy of EEG signals under successful stop versus failed stop conditions. The four kinds of classifiers were as follows:

#### 2.7.1. K-Nearest-Neighbor Classifier (KNNC)

The KNNC is the k-nearest-neighbor classifier and is one of the basic classifiers for pattern recognition. The principle of this method is an intuitive concept that data instances of the same class should be closer in the feature space. For a given data point x of one unknown class, the distance between x and all the data points in the training data was investigated, and the class of X was determined by K, i.e., the nearest point of X. Due to its simplicity, KNNC is frequently used as a base method in comparison with other sophisticated methods in pattern recognition. KNNC regulation is well known in the pattern recognition literature [27,28]. 

#### 2.7.2. Parzen Density-Based Classifier (PARZENDC)

The PARZENDC is a technique for nonparametric density estimation. It is based on a PARZEN kernel density estimate with its smoothing kernel parameter acquired as a maximum likelihood estimate [29]. Using a given kernel function, the technique approximates a given training set distribution via a linear combination of kernels centered on the observed points. In this work, we separately approximate densities for each of the two classes and assign a test point to the class with maximal posterior probability [30]. The kernel width is optimized for the training set using a leave-one-out error estimation [30]. 

#### 2.7.3. Linear Discriminant Analysis (LDA) Classifier

Linear discriminant analysis was originally developed by [31], and it is the standard method for classification. This method often produces models as accurate as of more complex classifier. LDC analysis can be used only for classification not for regression. The target variable may have two or more categories. Moreover, for a two class problem, one canonical discriminant function can be constructed for classification of the two groups of cases. The discriminant function is formulated by a linear combination of the feature variables:(1)$$D=\;a_{0}+\sum_{i=1}a_{i}x_{i}$$

Assume *n* is the number of feature variables, the $x_{i}$ are the values of the feature variables, and the $a_{i}$ are coefficients estimated from the input data during training so that the separation between the distributions of the discriminant scores, D, of the two groups is a maximum. This is accomplished by maximization of the ratio of the between-groups sum of squares to the within-groups sum of squares for the two distributions of the discriminant scores [32].

#### 2.7.4. Quadratic Discriminant Analysis (QDA) Classifier

The quadratic classifier was used in machine learning and statistical classification to separate measurements of two or more classes of objects or events by a quadric surface. It is a more general version of the linear classifier. Here, we used a quadratic classifier for our recognition purpose based on literature review [33]. This classifier gives better results than the other classifiers. In particular, a Pattern Recognition Tools (PRTools) function called PARZENDC, KNNC, LDC, and QDC were employed for classifier design.

### 2.8. EEG Features Extraction and Selection Method

The EEG signal epoch was extracted from 1 to 500 ms to measure power spectral density (PSD) with inter-subject (between-subjects) and intra-subject (within-subjects) variability. The EEG signals from 1 to 50 Hz frequency were used to observe the power spectrum density under successful stop and failed stop trials. The calculated PSDs of the EEG signal during the successful stop and failed stop trials were used as the input feature for the LDA, QDA, PARZEN, and KNNC classifiers. In the feature extraction process, the PSDs of EEG signals were investigated in eight EEG-channels, which included F3 and F4 (frontal lobe), C3 and C4 (motor cortex), P3 and P4 (parietal lobe) and C3 and C4 (occipital lobe) under inter-subject and intra-subject variability. Most previous studies used power spectral density as a feature to classify EEG signals [34,35,36]. Consequently, in this study, we propose for the first time a classification system to predict successful stop versus failed stop under battleground scenario using power spectral density as input features. We used both parametric (QDA and LDA) and nonparametric (KNNC and PARZEN) algorithms to predict the response inhibition function.

In present work, we used the average power spectral density of EEG signals as input features to the PRTools—pattern recognition tools function from 1 to 500 ms using forward feature selection method during successful stop and failed stop trials. The forward feature selection technique begun with a blank set, X = 0, to which the most significant features with respect to X were added. Forward feature selection was utilized to select the best power spectral features to predict the successful stop and failed stop. The forward feature selection technique was started by investigating all subsets of features that consist of a single input attribute. In other words, we begun by measuring the leave-one-out cross validation (LOOCV) error of the one component subsets, {X1}, {X2}, {XM}, where M is the input dimensionality. Consequently, we examined the best individual feature, X (1). Moreover, the feature selection technique is called variable selection process. Feature selection denotes to the selection of a subset of related features used for making a classification system. Feature selection process was utilized for three main reasons: (I) generalization of system to make them easier to understand, (II) the shortening of classification model training times, and (III) the improvement of generalization by reducing overfitting [37].

Moreover, in this study, we used leave-one-out cross validation (LOOCV) for training and testing datasets in which one part for testing and remaining for training with equal sampling for successful and unsuccessful trials were used. Cross validation is a statistical method to evaluate and compare machine learning algorithms via dividing the EEG data into two parts: one utilized to train a classification model and the other utilized to validate the classification model. The accuracy of the suggested classification model was investigated by leave-one-out cross validation method [38]. This technique reduces the probability of obtaining erroneous outcomes, as it studies multiple splits of the EEG signals.

### 2.9. Statistical Analysis

In ERP analysis, the yellow asterisks show pairwise significance (*p* < 0.01) difference using Wilcoxon signed-rank test between the go and successful stop epochs. Green asterisks display pairwise significance (*p* < 0.01) difference in the go and failed stop epochs. Violet asterisks show pairwise significance (*p* < 0.01) difference using the Wilcoxon signed-rank test between the successful stop and failed stop epochs. Additionally, in ERSP analysis, the statistical significant (*p* < 0.01) difference with multiple comparison was measured using the default setting in EEG laboratory toolbox [10]. In power spectrum density (PSD) analysis, the statistical significance (*p* < 0.05) difference of power spectrum between successful stop and failed stop epochs was observed with Wilcoxon signed-rank test in F3 and F4 (frontal lobe), C3 and C4 (motor cortex), P3 and P4 (parietal lobe), and C3 and C4 (occipital lobe) under inter-subject and intra-subject variability. In addition, the pairwise significance difference (*p* < 0.05) was measured in *t*-test between the performance of QDA, LDA, KNNC, and PARZEN classifiers under inter-subject variability.

## 3. Results

### 3.1. Behavioral Results

Table 1 shows the behavioral results during go trial and stop trial with inter-subject and intra-subject variability. After removing the artifact trials, the number of trials per condition was measured for the behavioral analysis of the successful go 2522 trials, successful-stop 424 trails, and failed-stop 592 trails. In go trial, average reaction time (RT) was observed (375 ± 30 ms), and successful go (SG) ratio was measured (91% ± 4%). In stop trail, average stop signal delay (SSD) was investigated (195 ± 36 ms), stop-signal reaction time (SSRT) was observed (180 ± 41 ms), and successful stop (SS) ratio was investigated (43% ± 12%). In addition, the averaged inhibition function approached around 50% at SSD and error rate level increased with the length of SSD. Asterisks display the significant difference in one-way ANOVA between the RT, SSD, and SSRT (F (2, 57) = 167.5, *p* < 0.01])

### 3.2. Event-Related Potential (ERP) Results

Figure 5 shows the average event-related potential (ERP) under go, successful stop, and failed stop trials at F3, F4, and C3 channels. In this study, we used the default setting in EEG laboratory toolbox for generating the ERP image [10]. It considers a moving window of 10-epoch size with 50% overlap. The yellow asterisks indicate pairwise significance (*p* < 0.01) in Wilcoxon signed-rank test between the go and successful stop conditions. Green asterisks show pairwise significance (*p* < 0.01) difference in the go and failed stop conditions. Violet asterisks show pairwise significance (*p* < 0.01) difference by Wilcoxon signed-rank test between the successful and failed stop condition. The human inhibition-related neural markers ERP N200 and P300 peaks were observed to be increased in successful stop than in go and failed stop conditions over the frontal cortex (F3 and F4 channels) and the presupplementary motor area (C3 channels).

### 3.3. Event-Related Spectral Perturbation (ERSP) Results

Figure 6 shows the average event-related spectral perturbation (ERSP) of the frontal cortex during go, successful stop, and failed stop trials at F3, F4, and C3 channels. The response inhibition-related EEG activity of delta (1–4 Hz) and theta (4–7 Hz) powers were investigated increased in successful stop than in go and failed stop conditions at the frontal cortex (F3 and F4) and the presupplementary motor area (C3).

Figure 7 displays the average ERSP of the EEG signals under the “Successful stop—Go” and “Failed stop—Go” conditions at frontal cortex (F3 and F4 channels) and supplementary motor area (C3 channel). The human inhibitory control-related EEG activity of delta (1–4 Hz) and theta (4–7 Hz) band powers were detected increased in “Successful stop—Go” condition at F3, F4, and C3 channels. However, we observed greater EEG activity of delta (1–4 Hz) and theta (4–7 Hz) band powers in the “Failed stop—Go” than in the “Successful stop—Go” condition at F3, F4, and C3 channels.

### 3.4. Power Spectral Density (PSD) of Inter- and Intra-Subject Variability under Human Inhibition

The EEG signals were extracted from 1 to 500 ms after the onset of the stimuli, to investigate the average PSD of EEG signals (1-50Hz) during successful stop and failed stop conditions. In this study, we used the EEG frequency spectrum (1–50 Hz) (i.e., PSD) as input in the machine learning approaches with inter- and intra-subject variability. Figure 8 shows the average PSD of all subjects (i.e., inter-subject variability) in human inhibition-related brain regions included frontal lobe (F3 and F4), motor cortex (C3 and C4), parietal lobe (P3 and P4), and occipital lobe (O1 and O2) under successful stop and failed stop trials. Asterisk display significant difference between successful stop trials and failed stop trials by Wilcoxon signed-rank test (*p* < 0.05). We observed the delta (1–4 Hz) and theta (4–7 Hz) band powers slightly increased in failed stop trials than in successful stop trials in F3, F4, C3, C4, P3, P4, O1, and O2 EEG channels. We found no significant difference in PSD of P3, P4, O1, and O2 EEG channels. These brain activities changes under human response inhibition revealed the inter- and intra-subject variability. In addition, the intra-subject variability in EEG power spectral density of each subject, from 1–20, is presented in the Appendix A.

### 3.5. Classification System Performance using the EEG Frequency Spectrum (1–50 Hz) as Input in the Machine Learning Approaches with Inter- and Intra-Subject Variability

In present study, a classification model was developed to predict the neural activities of human inhibition using inter-subject and intra-subject variability. In this work, simple nonparametric and parametric classifiers were utilized to estimate model performance, because these classifiers are very easy to use. If these simple classifiers achieve batter performance in highest accuracy than the use of more difficult classifiers algorithms can be escaped. Moreover, the commonly parametric machine learning algorithms were used including the quadratic discriminant analysis (QDA) classifier and the linear discriminant analysis (LDA) classifier. The nonparametric machine learning algorithms were used including the K-nearest neighbor classifier (KNNC) and Parzen density-based classifier (PARZEN). The nonparametric (KNNC and PARZEN) and parametric (QDA and LDA) classification algorithms were utilized to classify successful stop and failed stop (FS) trials. We used the average power spectral density (PSD) of EEG signals (1–50 Hz frequency) as input features for classifiers from 1 to 500 ms using forward feature selection method during successful stop and failed stop trials, because the inhibition-related ERP N200 and P300 peaks were observed from 1 to 500 ms after stimulus onset at F3, F4, and C3 channels. In the ERSP analysis, we observed that the delta (1–4 Hz) and theta (4–7 Hz) power increased after stimulus onset from 1 to 500 ms at F3, F4, and C3 channels.

The accuracy of the classification system was observed during the successful stop and failed stop trials with inter- and intra-subject. Table 2 shows the comparison of the variability between inter-subject and intra-subject of each participant’s performance with accuracy (%) in F3 and F4 (frontal cortex). We found the classification accuracy of 92.85% with QDA in intra-subject, and the classification accuracy was reduced to 71.66% with KNNC in inter-subject at F3 (i.e., left frontal cortex). However, we observed highest classification accuracy of 95.31% with QDA in intra-subject, and the classification accuracy was found to be 69.54% with KNNC in inter-subject at F4 (i.e., right frontal cortex).

Table 3 displays the evaluation of the variability between inter-subject and intra-subject of each participant’s performance with accuracy in C3 and C4 (motor cortex). We saw the best classification accuracy of 97.61% with QDA in intra-subject, and major drop in the classification accuracy was found as 70.91% with QDA in inter-subject at C3 (left motor cortex). Moreover, we found classification accuracy of 93.75% with QDA in intra-subject, and the classification accuracy was investigated to be 69.77% with KNNC in inter-subject at C4 (right motor cortex).

Table 4 shows the comparison of the variability between inter-subject and intra-subject of individually subject performance with accuracy in P3 and P4 (parietal lobe). We observed classification accuracy of 95.23% with QDA in intra-subject, and the classification accuracy was obtained as 70.05% with QDA in inter-subject at P3 (left parietal lobe). Furthermore, we investigated best classification accuracy of 96.87% with QDA in intra-subject, and the classification accuracy was found to be 70.01% with QDA in inter-subject at P4 (right parietal lobe).

Table 5 displays the comparison of the variability between inter-subject and intra-subject of individually subject performance with accuracy in O1 and O2 (occipital lobe). We found the best classification accuracy of 97.61% with QDA in intra-subject, and the classification accuracy was obtained as 69.57% with QDA in inter-subject at O1 (left occipital lobe). Furthermore, we investigated classification accuracy of 93.75% with QDA in intra-subject, and the classification accuracy was found to be 70.05% with PARZEN in inter-subject at O2 (right occipital lobe). These EEG findings show how the variability between inter-subject and intra-subject affects the performance of the classification system.

### 3.6. Comparison of Classification Model Performance between Inter-Subject and Intra-Subject Variability using the EEG Frequency Spectrum (1–50 Hz) as Input in the Machine Learning Approaches

This section presents the classification system performance with QDA, LDA (parametric), KNNC and PARZEN (nonparametric) classifier using the EEG frequency spectrum (1–50 Hz) as input in the machine learning approaches. The accuracy of the present classification model was investigated under successful stop and failed stop (FS) trials. Figure 9 shows the comparison of inter- and intra-subject variability of participant’s maximum performance in accuracy under successful stop and failed stop trials. Asterisks show pairwise significance difference (* *p* < 0.05) in *t*-test between the QDA, LDA, KNNC, and PARZEN. In this model, the highest classification accuracy of 97.61% was achieved with QDC in the left motor cortex (C3) under intra-subject variability. However, we found classification accuracy of 71.66% was reached with KNNC in the left motor cortex (C3) under inter-subject variability. These results clearly demonstrate that how the variability between inter-subject and intra-subject affects the performance of the classification model.

## 4. Discussion

In this study, a classification model was developed to predict the response inhibition of subjects based on their EEG signals under the battleground scenario. The ERP and ERSP results were used as a neural marker for inhibition of the human response. In addition, the PSD of EEG signals at F3, F4, C3, C4, P3, P4, O1, and O2 channels were investigated under successful stop versus failed stop condition with inter-subject versus intra-subject variability. The PSD of EEG signals from (1–50 Hz) was utilized as the input feature for the parametric (QDA and LDA) and nonparametric (KNNC and PARZEN) machine learning approaches. The classification results reveal that the proposed method prediction accuracy of 97.61% was achieved with QDC classifier under intra-subject variability at C3 channel. However, major drop in the classification accuracy was found as 71.66% with KNNC in inter-subject variability at F3 channel. Previous studies reported that the frontal and presupplementary motor areas of the brain have been related to human response inhibition [13,19]. This study show that how inter-subject and intra-subject variability affects the performance of the classification system under inhibition.

### 4.1. Neural Activity Change with Inter-Subject and Intra-Subject Variability

In our study, we observed inhibition-related biomarkers (i.e., ERP N200 and P300 waves). These results are similar to preceding studies that investigated the frontal lobe (i.e., F3 and F4 channels) and presupplementary motor area (i.e., C3 channel) of the brain, which are associated with human response inhibition [3,4,5,6,7,8,9]. In addition, the ERSP of delta and theta band powers were increased after successful stop at F3, F4, and C3 channels.

Moreover, the previous studies of functional connectivity with inter-subject and intra-subject variability reported that EEG activity of brain regions was observed to be dissimilar in both conditions [11,12]. They observed greater intra-subject variability than in inter-subject variability using functional connectivity in occipital lobe of the brain [11]. In our study, we also explored the similar EEG activities changes under human response inhibition over the frontal cortex (F3 and F4) and presupplementary motor area (C3) of the brain. Therefore, inter- and intra-subject variability shows the different result of the classification model. Most of the former studies performed the average analysis of all participants. Therefore, for the first time in this study, we investigated inter-subject and intra-subject variability under human inhibitory control. These results can be used effectively in clinical research to improve inhibition-related mental disorders, like ADHD, OCD, and schizophrenia.

### 4.2. Relationship between Inter- and Intra-Subject Variability

For the first time, our study has explored the specific relationship between inter-subject and intra-subject variability under the battleground scenario. We observed the relationship between human cognitive inhibition and shooting the target, which can offer numerous practical benefits in military as well as in clinical research, such as to improve the psychopathology of ADHD, OCD, and schizophrenia. First, the response inhibition relations with cognitive abilities could help us to identify the EEG neural markers (ERP, ERSP, and PSD) of the individual participants (i.e., healthy subject or patient) using battlefield scenario. Second, we can easily eliminate errors in classification model using inter- and intra-subject variability. These results increase the evidence that the battleground scenario can be used to improve the cognitive ability of patients.

Cognitive inhibition training improved the control of response inhibition ability, which may reduce the error rate for stopping the brain–computer interface (BCI) system [39,40,41,42]. The present study provides an initial view of this relationship by comparing the performance of normal subjects during inter- and intra-subject variability in battleground scenario. However, previous studies have shown that inhibitory control could be trained by stop signal task [43,44]. Present study demonstrates through experimental results that the PSD with the QDA classifier can provide highest classification accuracy of 97.61% than the other commonly used features in machine learning. For the first time, this study presents the highest accuracy of 97.61% to classify complex mental states related to human response inhibition under the battleground scenario. In addition, these results show that inter- and intra-subject variability affects the performance of the classification system due to the neurophysiological state changes of the individual participants.

Previous studies reported that the correct classification of ERP waves in EEG signals is a difficult method that requires efficient signal processing and machine learning techniques. ERP analysis provides non-invasive measurements of electrical activity in the human brain. It is associated with the response to stimuli [45,46,47]. The previous study yielded a higher classification accuracy of 93.75% using ERP input features in machine learning techniques [45,46,47]. In our article, using PSD with the QDA classifier achieved the highest classification precision of 97.61% than in ERP features commonly used in former machine learning algorithms. The power spectrum of the EEG signal increased the performance of the classification system.

### 4.3. Application and Future Works

In addition, the current classification model can be used to predict EEG neural activity of human inhibition-related disorders, like ADHD, obsessive-compulsive disorder (OCD), and schizophrenia. For example, when a participant (ADHD, OCD, and schizophrenia patient) begins preparing to response the target image, the power spectral density of the EEG signals from 1 s or 500 ms in the brain region related to the inhibition of response will be investigated, such as in the frontal lobe and the presupplementary motor area [13,19,48,49,50,51,52]. After that, the EEG power spectrum of the frontal lobe and the presupplementary area will be transferred to the proposed classification model to predict the performance of the participant in real time. If the participant’s performance is predicted as a success, then the classification system will produce a tone or signal to encourage the patient to response the target and inhibit response with nontarget. The 500 ms EEG signals can be analyzed by this procedure every few milliseconds until the patient improve performance. However, it is a challenging task for future work because in a real environment, the neurophysiological state changes (i.e., fatigue, stress, and mind-wandering) affect the EEG signal of patients. This novel study of inhibition can be used effectively to improve the psychopathology of ADHD, OCD, and schizophrenia.

The limitation is that we used 20 subjects. For future work, the number of subjects will be increased to validate the design of the experiment for a large dataset. In addition, we will perform the brain connectivity analysis with a large dataset. We will use deep learning-based models to analyze this dataset and develop a real-time model for the inhibition of human response with a high rate of accuracy. In addition, we will compare the subject’s performance with ICA and without ICA analysis during inter-subject and intra-subject variability.

## 5. Conclusions

To conclude, based on the results of the classification, we found differences in performance between inter- and intra-subject variability during successful stop versus failed stop trials. The highest classification accuracy observed from individual participants (intra-subject variability) was 97.61% with QDA, although average participants (inter-subject variability) obtained highest classification accuracy of 71.66% with KNNC. We observed reduced performance of the classification system accuracy of 71.66% with KNNC in inter-subject variability. These results show that the mental status of the individual participants affects the performance of the classification system. The results of the present study suggest that this classification system can be applied effectively to improve the psychopathology of attention deficit hyperactivity disorder, obsessive-compulsive disorder, schizophrenia, and suicidality patient by training their mental state under the battlefield scenario. For future work, the number of subjects will be increased in order to obtain more clean and substantial EEG dataset for the accuracy of the mental state classification.

## Figures and Tables

**Figure 1 brainsci-10-00726-f001:**
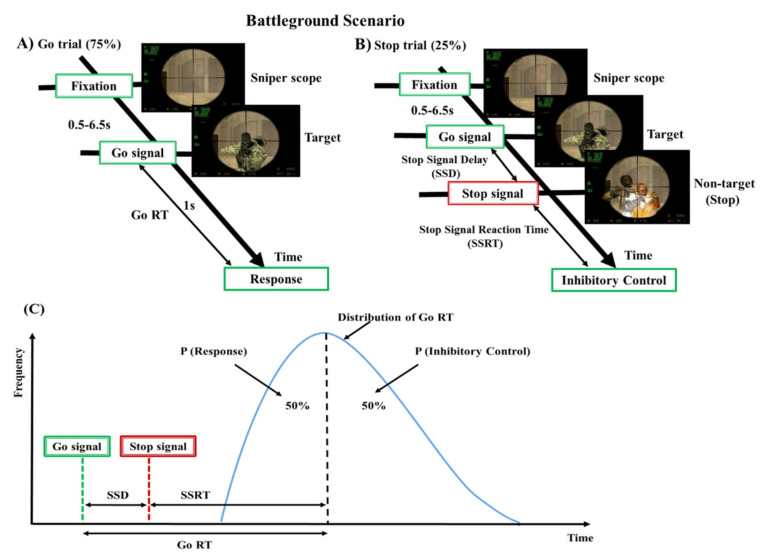
Design of stop signal task: (**A**) go trial and (**B**) stop trial under battleground scenario. Each subject performed go trials (75%) and stop trials (25%) according to the stimuli presented in the battleground scenario. (**C**) The theoretical model of the stop-signal task and where P is the probability of responding to the stop-signal.

**Figure 2 brainsci-10-00726-f002:**
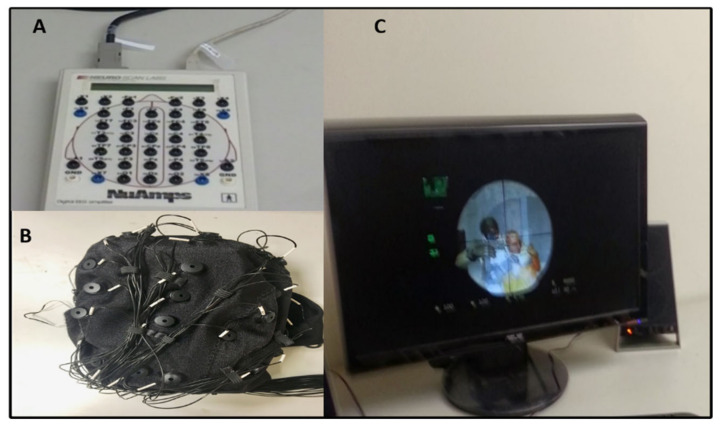
The electroencephalography (EEG) devices used in battleground scenario. (**A**) Neuro Scan NuAmps signal amplifier. (**B**) EEG cap with 32-channels. (**C**) Experimental screen in battleground scenario.

**Figure 3 brainsci-10-00726-f003:**
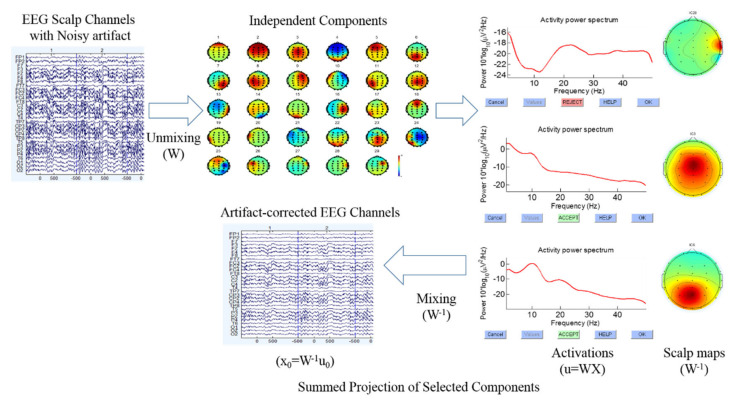
Flowchart of the independent component analysis and back projecting the retained independent components to artifact free EEG channels (i.e., clean EEG signals).

**Figure 4 brainsci-10-00726-f004:**
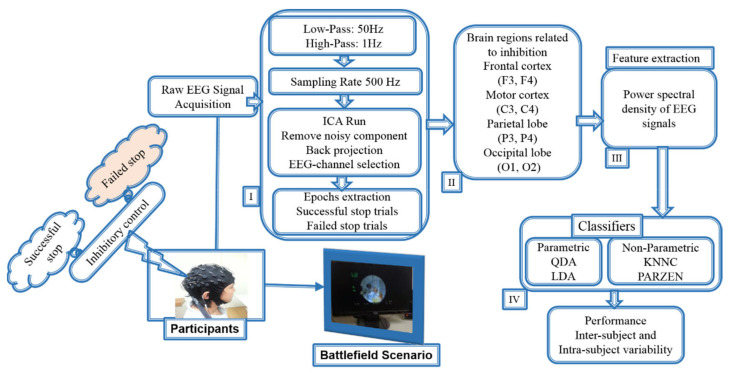
Flowchart of the classification system architecture. (**I**). Preprocessing steps of acquired EEG signals. (**II**). Human inhibition-related brain regions. (**III**). Analysis of EEG signals power spectral density (PSD) under inter-subject and intra-subject at preparation state before stimulus. (**IV**). The QDA and LDA (parametric) and KNNC and PARZEN (nonparametric) classifiers performance outcomes comparison during inter-subject and intra-subject variability.

**Figure 5 brainsci-10-00726-f005:**
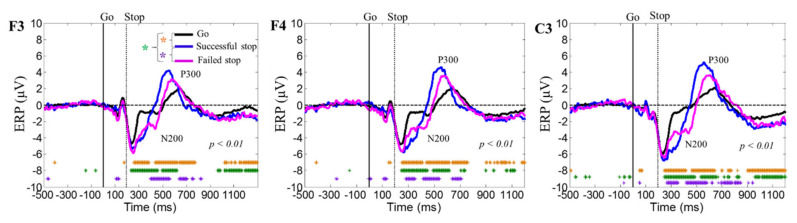
The average event-related potential (ERP) during go, successful stop, and failed stop trials at F3, F4, and C3 channels. Yellow asterisks indicate pairwise significance (*p* < 0.01) in Wilcoxon signed-rank test between the go and successful stop conditions. Green asterisks show pairwise significance (*p* < 0.01) in Wilcoxon signed-rank test between the go and failed stop. Violet asterisks show pairwise significance (*p* < 0.01) in Wilcoxon signed-rank test between the successful and failed stop.

**Figure 6 brainsci-10-00726-f006:**
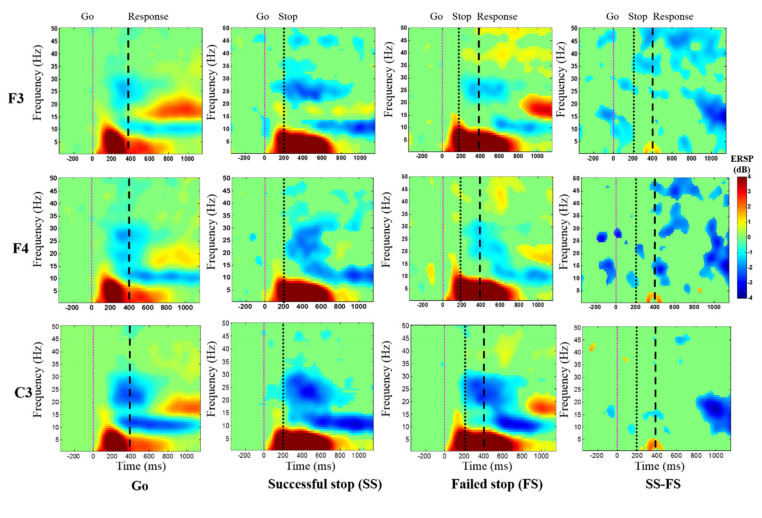
The average event-related spectral perturbation (ERSP) of the frontal cortex and the supplementary motor area of the brain during go, successful stop, and failed stop trials at F3, F4, and C3 channels. The first magenta line shows go-stimulus onset. The second black dashed line reveals stop signal onset. The third black dashed line presents response onset. Statistically significant at *p* < 0.01. Color bars show the scale of ERSP.

**Figure 7 brainsci-10-00726-f007:**
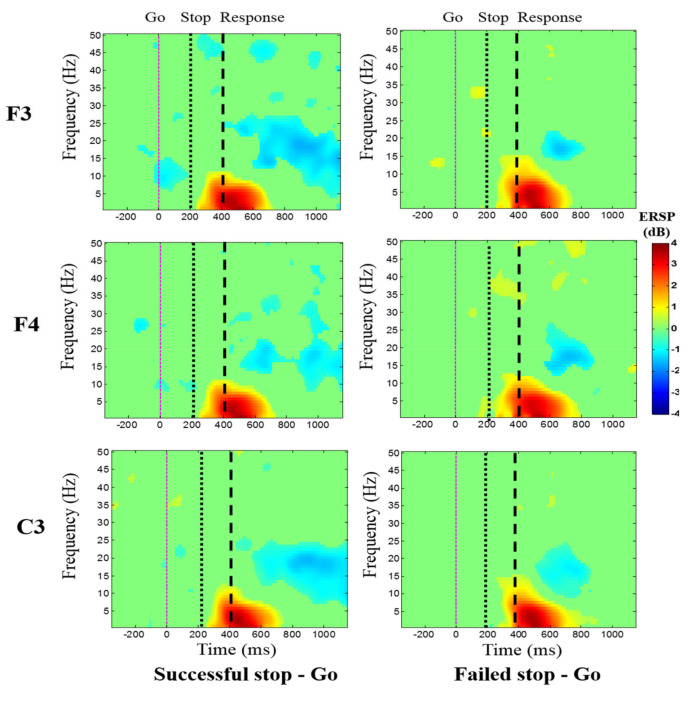
The average ERSP of the EEG signals under “Successful stop—Go” and “Failed stop—Go” conditions at F3, F4, and C3 channels. The first magenta line shows go-stimulus onset. The second black dashed line reveals stop signal onset. The third black dashed line presents response onset. Statistically significant at *p* < 0.01. Color bars show the scale of ERSP.

**Figure 8 brainsci-10-00726-f008:**
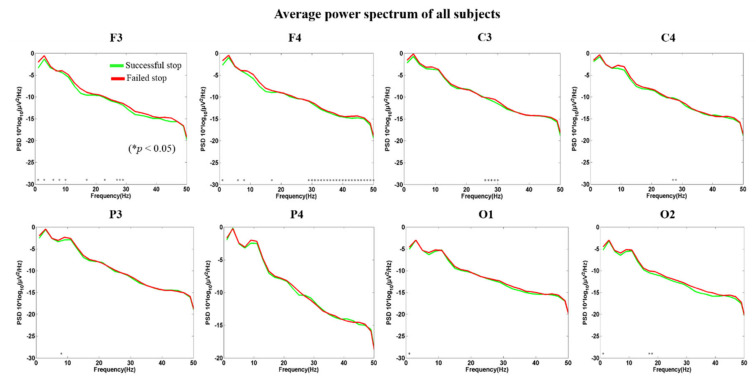
The average power spectral density (PSD) of all subjects (inter-subject variability) at F3, F4, C3, C4, P3, P4, O1, and O2 under successful stop and failed stop trials. Asterisk show significant difference between successful stop trials and failed stop trials by Wilcoxon signed-rank test (at **p* < 0.05).

**Figure 9 brainsci-10-00726-f009:**
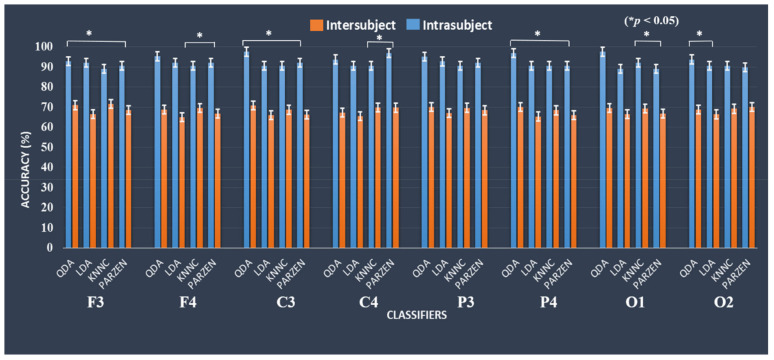
The QDA and LDA (parametric) and KNNC and PARZEN (nonparametric) classifiers performance outcomes comparison in accuracy and standard error bars during inter-subject and intra-subject variability at F3, F4, C3, C4, P3, P4, O1, and O2. Asterisks show pairwise significance difference (* *p* < 0.05) in *t*-test between the QDA, LDA, KNNC, and PARZEN.

**Table 1 brainsci-10-00726-t001:** Behavioral results during go trial and stop trial with inter-subject and intra-subject variability. Standard deviation (SD). Asterisks show the significant difference in one-way ANOVA between the reaction time (RT), stop signal delay (SSD), and stop signal reaction time (SSRT) (F (2, 57) = 167.5, *p* < 0.01).

Subject	Go Trial (75%)		Stop Trial (25%)		
ID	RT (ms)	SG Ratio (%)	SSD (ms)	SSRT (ms)	SS Ratio (%)
1	446	94	183	263	48
2	376	99	172	204	49
3	381	95	167	214	51
4	346	93	167	179	29
5	378	90	260	118	54
6	416	88	198	218	54
7	382	84	184	198	48
8	350	98	167	183	27
9	366	86	166	200	38
10	420	94	167	253	10
11	366	82	215	151	49
12	346	93	219	127	49
13	338	96	167	171	33
14	350	95	166	184	49
15	352	94	234	118	52
16	336	87	193	143	51
17	382	89	167	215	46
18	434	85	290	144	57
19	376	93	249	127	35
20	368	92	167	201	21
(Average ± SD)	375 ± 30 *	91 ± 4	195 ± 36 *	180 ± 41 *	43 ± 12

**Table 2 brainsci-10-00726-t002:** Result of leave-one-out cross validation using the EEG frequency spectrum (1–50 Hz) as input in the machine learning approaches including quadratic discriminant analysis (QDA), linear discriminant analysis (LDA) (parametric), K-nearest neighbor classifier (KNNC) and Parzen density-based (PARZEN) (nonparametric). The performance of the classification system is compared in accuracy (%) during inter-subject and intra-subject variability at frontal lobe (F3 and F4) of the brain.

Subject	F3				F4			
ID	QDA	LDA	KNNC	PARZEN	QDA	LDA	KNNC	PARZEN
1	73.46	65.30	77.55	65.30	87.75	57.14	55.10	63.26
2	67.21	65.57	68.85	60.65	49.18	63.93	62.29	63.93
3	58.82	54.90	60.78	52.94	60.78	58.82	62.74	56.86
4	73.07	73.00	73.00	75.00	73.07	73.00	86.53	71.15
5	72.00	72.00	68.00	74.00	50.00	64.00	70.00	62.00
6	75.60	65.85	78.04	70.73	56.09	65.85	51.21	70.73
7	64.91	63.15	56.14	63.15	52.63	66.66	66.67	64.91
8	73.46	73.47	71.42	73.46	75.51	73.46	87.75	75.52
9	68.51	62.96	61.11	66.66	66.67	62.96	72.22	64.81
10	90.62	92.18	89.06	90.63	95.31	92.19	90.62	92.19
11	59.25	59.26	57.40	62.96	85.18	51.85	59.25	61.11
12	65.21	67.39	65.21	56.52	73.91	58.69	60.86	60.87
13	79.06	72.09	83.72	74.41	67.44	67.40	69.76	67.43
14	64.28	64.29	66.66	69.04	71.42	57.14	71.43	71.41
15	92.85	59.52	80.95	71.42	78.57	61.90	66.66	54.76
16	61.81	49.09	56.36	54.54	67.27	63.63	74.54	56.36
17	58.62	67.24	75.86	56.89	51.72	58.62	55.17	51.72
18	66.00	62.00	80.00	74.00	58.00	58.10	77.00	72.00
19	73.46	67.34	77.55	79.59	75.51	65.30	71.42	75.52
20	81.63	75.51	85.71	79.59	81.64	79.59	79.58	79.59
Accuracy (Avg. ± SD)	70.99 ± 9.40	66.60 ± 8.61	71.66 ± 9.81	68.57 ± 9.38	68.88 ± 12.98	65.01 ± 8.88	69.54 ± 10.74	66.80 ± 9.33

**Table 3 brainsci-10-00726-t003:** Result of leave-one-out cross validation using the EEG frequency spectrum (1–50 Hz) as input in the machine learning approaches including QDA, LDA (parametric), KNNC and PARZEN (nonparametric). The performance of the classification system is compared in accuracy (%) during inter-subject and intra-subject variability at motor cortex (C3 and C4) of the brain.

Subject	C3				C4			
ID	QDA	LDA	KNNC	PARZEN	QDA	LDA	KNNC	PARZEN
1	77.55	73.46	73.47	73.46	63.26	63.27	65.30	71.42
2	77.04	65.57	63.93	67.21	49.18	55.73	63.93	59.01
3	66.66	54.90	66.66	66.67	64.70	64.71	68.62	72.54
4	73.07	73.00	69.23	71.15	75.00	73.07	84.61	76.92
5	76.00	58.00	56.00	60.00	68.00	66.00	70.00	66.00
6	58.53	58.54	56.09	53.65	48.78	56.09	60.97	51.21
7	57.89	70.17	59.64	70.18	66.66	63.15	68.42	64.91
8	69.38	73.46	85.71	73.46	73.46	77.55	81.63	77.56
9	72.22	68.51	75.92	72.22	75.92	64.81	59.25	74.07
10	90.63	90.62	90.63	92.18	93.75	90.62	90.60	96.87
11	90.74	62.96	64.81	55.55	55.56	62.96	57.40	57.41
12	71.73	60.86	76.08	60.86	65.21	54.34	54.35	56.52
13	67.44	69.76	62.79	72.09	69.76	67.44	81.39	79.06
14	57.14	57.15	69.04	52.38	71.42	57.14	78.57	69.04
15	97.61	57.14	71.42	61.90	92.85	59.52	71.42	69.04
16	61.81	58.18	54.54	58.18	54.54	74.54	69.09	76.36
17	50.00	65.51	65.51	60.34	60.34	62.06	56.89	51.72
18	58.00	60.00	66.00	60.00	52.00	54.00	58.00	74.00
19	63.26	63.27	67.34	63.26	67.34	63.26	69.38	65.30
20	81.63	79.59	79.60	81.64	79.59	79.60	85.71	85.72
Accuracy (Avg. ± SD)	70.91 ± 12.27	66.03 ± 8.74	68.72 ± 9.28	66.31 ± 9.55	67.36 ± 12.20	65.49 ± 9.21	69.77 ± 10.50	69.73 ± 11.06

**Table 4 brainsci-10-00726-t004:** Result of leave-one-out cross validation using the EEG frequency spectrum (1–50 Hz) as input in the machine learning approaches including QDA, LDA (parametric), KNNC and PARZEN (nonparametric). The performance of the classification system is compared in accuracy (%) during inter-subject and intra-subject variability in parietal lobe (P3 and P4) of the brain.

Subject	P3				P4			
ID	QDA	LDA	KNNC	PARZEN	QDA	LDA	KNNC	PARZEN
1	73.46	71.42	61.22	61.23	59.18	73.46	65.30	69.38
2	62.29	65.57	72.13	63.93	65.57	72.13	60.65	68.85
3	64.70	56.86	62.74	56.86	70.58	58.82	66.66	66.67
4	76.92	73.07	78.84	76.92	80.76	73.07	78.84	75.00
5	58.00	54.00	72.00	62.00	52.00	52.00	72.00	58.00
6	60.97	53.65	73.17	65.85	51.21	48.78	73.17	68.29
7	75.43	77.19	71.92	77.19	57.89	61.40	68.42	57.89
8	77.55	75.51	79.59	85.71	73.46	73.47	79.59	77.55
9	66.66	68.51	68.52	68.51	66.67	61.11	66.66	70.37
10	93.75	92.18	90.62	92.19	96.87	90.62	90.61	90.62
11	77.77	51.85	68.51	62.96	79.62	62.96	53.70	55.55
12	65.21	73.91	67.39	63.04	93.47	58.69	69.56	67.39
13	72.09	74.41	72.00	72.09	72.09	79.06	76.74	74.41
14	57.14	61.90	59.52	59.53	54.76	54.76	66.66	61.90
15	95.23	59.52	64.28	66.67	92.85	61.90	54.76	57.14
16	70.90	67.27	69.09	58.18	63.63	58.18	61.81	45.45
17	50.00	55.17	44.82	56.89	58.62	62.06	56.89	55.17
18	54.00	64.00	64.00	66.00	62.00	62.00	62.00	60.00
19	61.22	63.26	75.51	71.42	61.22	63.26	65.30	59.18
20	87.75	81.63	77.55	81.63	87.75	79.59	79.59	81.63
Accuracy (Avg. ± SD)	70.05 ± 12.19	67.04 ± 10.27	69.67 ± 9.11	68.44 ± 9.61	70.01 ± 13.87	65.36 ± 10.11	68.44 ± 9.12	66.02 ± 10.33

**Table 5 brainsci-10-00726-t005:** Result of leave-one-out cross validation using the EEG frequency spectrum (1–50 Hz) as input in the machine learning approaches including QDA, LDA (parametric), KNNC and PARZEN (nonparametric). The performance of the classification system is compared in accuracy (%) during inter-subject and intra-subject variability in occipital lobe (O1 and O2) of the brain.

Subject	O1				O2			
ID	QDA	LDA	KNNC	PARZEN	QDA	LDA	KNNC	PARZEN
1	61.22	63.26	59.18	61.22	65.30	69.38	67.34	67.35
2	59.01	59.00	50.18	59.01	55.73	54.09	47.54	60.65
3	64.70	60.78	66.66	62.74	54.90	54.90	60.78	62.74
4	69.23	71.15	75.00	75.00	73.07	73.07	80.76	84.61
5	58.00	62.00	62.00	50.00	62.00	68.00	72.00	64.00
6	63.41	63.41	70.73	63.41	73.17	60.97	73.17	60.97
7	64.91	59.64	57.89	57.89	56.14	57.89	66.66	64.91
8	73.47	73.46	79.59	77.55	83.67	85.71	81.63	85.71
9	64.81	62.96	68.51	70.37	57.40	62.96	61.11	61.11
10	96.87	89.06	92.18	89.06	93.75	90.62	90.62	89.06
11	77.77	59.25	61.11	64.81	59.25	51.85	59.25	55.55
12	58.69	73.91	76.08	67.93	80.43	71.73	80.43	82.60
13	74.41	79.06	72.09	76.74	72.09	67.44	67.44	74.41
14	61.90	73.80	69.04	73.80	64.28	69.04	66.66	69.04
15	97.61	54.76	66.66	69.04	78.57	59.52	64.28	71.42
16	58.18	63.63	67.27	54.54	63.63	63.63	52.72	63.63
17	62.06	63.79	68.96	62.06	60.34	58.62	65.51	60.34
18	66.00	56.00	70.00	62.00	64.00	66.00	72.00	70.00
19	77.55	61.22	69.38	59.18	71.42	61.22	65.30	63.26
20	81.63	79.59	83.67	79.59	85.71	81.63	89.79	89.79
Accuracy (Avg. ± SD)	69.57 ± 11.49	66.48 ± 8.80	69.30 ± 9.11	66.79 ± 9.34	68.74 ± 10.86	66.41 ± 10.08	69.24 ± 10.90	70.05 ± 10.37

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
