# Peer review of "Prediction of Human Inhibition Brain Function with Inter-Subject and Intra-Subject Variability"

_brainsci, 2020, doi:10.3390/brainsci10100726_

Round 1

Reviewer 1 Report

Although the idea and general objective of the study could be interesting, the manuscript is so messy that the study is not understandable. The main concerns are the following:

  • The writing and the English are not good: there are so many typos and unfinished sentences, making really difficult to follow the manuscript. 
  • I do not understand what the Authors mean with intra and inter-subject variability in this context. In the Introduction they did not provide a good explanation of this point and the references they used to give examples are not related to this topic (i.e. Ref. 11). Moreover, in the methodology section, they did not go deeper on what kind of data they used to separate intra and inter-subject data.
  • Methodology regarding EEG acquisition, pre-processing and analyses is not enough detailed, and they are not following the standards used in EEG literature. It is not clear if the Authors record EOG simultaneously; if they remove the artifacted portions of the raw data by hand or through ICA (both are mentioned); what type of segmentation are they doing because they mention two different things in the manuscript (from -500 to 1300, and to 1ms to 500 ms); why they used three different methods to analyze EEG signal (ERP, ERSP, and PSD) if they are just introducing PSD in the classificatory analyses. Moreover, they present the results for ERPs and ERSP but these are not discussed. 
  • The power spectral analyses section in methodology is not detailed at all, and most of the important information is missing: which method did you use to calculate PSD, how did you calculate it, did you use any kind of normalization? Which bands did you use? Which were the bins used for defining the bands?
  • On the contrary, the section explaining the machine learning classifiers is too detailed when this is not necessary at all. I mean, readers need a detailed methodology about EEG acquisition and analyses to replicate the study but they can check relevant literature to know about the basis of the different types of classificatory methods.
  • Again in section 2.7 it is not explained which were the specific variables used for classification methods: PSD yes, but in which bands, how did you separate intra and inter-subject epochs, ….
  • Statistical analyses are so weird. Why did the Authors use pair comparisons with Wilcoxon test? There are have three conditions, the correct way is to use an ANOVA or if the data are not parametric Mann-Whitney tests!! In addition, I do not understand how did you perform the averages for ERP analyses, and finally you are not discussing these results. I do not understand why the Authors are not using this data to perform classificatory analyses; I do not think the PSD measures are the most correct way to classify the trials.
  • The behavioral data are presented but they are not compared. Is there any significant difference?
  • All the results regarding ERP and ERSP are messy. I do not understand which were the real statistical differences among the conditions.
  • Results on the classificatory analyses are not clear. The authors talk about accuracy percentages in just one subject for intra-subject variability, and they based all the discussion on this.
  • The conclusions are not sustained by data and finally, the whole integrity of the manuscript is jeopardized.

To sum up, overall the manuscript is not high-quality research, is not well-designed, organized and discussed. So, I highly recommend the Authors to carefully revise the whole paper, define a good set of objectives, state clear why this kind of analyses can be promising and improve the methodology and the general writing to make the ideas clear and understandable.

Author Response

Dear Editor,

We have revised the all statements of the manuscript carefully. The authors are grateful to the editor and reviewer for the constructive comments on this manuscript. We believe all of the valuable suggestions have been addressed in our response and in the revised manuscript.

Reviewer 2 Report

The paper deals with an interesting matter concerning neural bases of an important brain activity that pertaining the ability to rapidly stop an ongoing activity due to a  changing request by the environment.

The study explores this in a more realistic scenario, and this a point of strenght and novelty.

Experimental design and methods seem sound, but some aspects particulary those concerning statistics for predictions need a bit more explanation.

Another point is that concerning EEG recording: 32 channels are really too a few to explore in detail complex activity as that studied by authors. Indeed, connectivity, beyond power spectrum analysis would have added more relevant informations.

Another important concern is language. English is really poor. In many points through the ms the meaning of sentences is hampered by inacurity of grammar and syntax.

Author Response

(The authors gave the same response as above.)

Reviewer 3 Report

They authors present a study on comparison of participants for successful stop and failed stop using EEG-based approach. The between-subject and within-subject performances are compared.

Use deep learning-based models for analyzing EEG signals. Since some deep learning-based models show better performance than the classical machine learning-based methods, the accuracy of this study could be improved, when deep learning-based models are exploited.

What is the limitation of this study? The authors should frankly specify the limitation of this study and suggest a scheme to overcome the limitation.

One important thing that the authors should know is that the importance of abstract. In the abstract of this manuscript, there are so many grammatical errors. Some of them are listed below. Any paper with an abstract of so many grammatical failures are rejected by any journals.

============================

L14-15: To explore 14 response inhibition in realistic environmental scenario. (Non sentence)

L21: during the successful stop (response inhibition) and failed stop (response) trials. --> the shoud be omitted

L23: affects --> affect

L27: and K-nearest neighbor classifier (KNNC), parzen density-based (PARZEN) (non-parametric). --> K-nearest neighbor classifier (KNNC), and parzen density-based (PARZEN) (non-parametric).

L28: The EEG activities were found increase --> found to increase

Author Response

(The authors gave the same response as above.)

Round 2

Reviewer 1 Report

I appreciate the Author’s responses and the effort to improve the manuscript. However, I don’t feel the article is ready for publication, and given this was the second version I doubt this manuscript has the potential to be published.

Although the spelling and the writing have improved a little bit, the quality of the writing is not ok, and there are still English typos (i.e. batter instead of better, page 3 line 106), so I do not consider the paper had been checked by a native English speaker.

The organization of ideas and sections is not clear, so the manuscript is still not understood. I appreciate the attempt of the Authors to clarify some aspects in the Introduction, such as the exact meaning of intra-subject variability in the context of the present work, but it is still unclear. For example, this paragraph taken from the manuscript means anything to me:

“Therefore, inter and intra-subject variability shows a different effect on the performance of the subjects. Most of the previous EEG studies reported averaged subject’s classification analysis 77 [7,8,9]. Consequently, we explore the classification model of inter-subject and intra-subject variability 78 under human inhibitory control.”

Then in the methodology, the authors added “inter-subject (between-subjects) and intra-subject (within-subjects) variability”. Obviously, I do know what inter and intra mean, my questions regarding this point were oriented to make the Authors explain what kind of measures they included for measuring intra-subject variability. They cannot talk about this given that they just have one measure per subject, there is no variability. In the paper the Author’s mentioned to exemplify the importance of intra-individual measures, Lauman et al. (2015) stated “Here, we characterized the brain organization of a single individual repeatedly measured over more than a year”. This is indeed a measure of intra-subject variability not what the Authors did in the present work, i.e. take just one measure and compare two conditions. I consider that is not correct, you are not talking about variability but about individual analysis.

Regarding methodology, changes made by the Authors are not enough for me. Again, they did not follow the common flow of steps in ERP/EEG literature. Some important steps are missing (baseline correction, epoch rejection…). It is not clear when the segmentation of the raw data was done. I recommend the Authors to take several ERP/EEG papers and try to describe the method in this way to make the procedure fully replicable.

The description of power spectral analyses is still deficient. Authors shouldn’t refer to Matlab or EEGlab websites to explain how to calculate PSD. They must give a detailed explanation (i.e. Welch method, kind of window, …). In fact, the methods used by Matlab (periodogram) and EEGlab (spectopo) are different, so the calculation of the power is different, so the results will be different. Which of them did you use to build your data matrix? Your answer tells me you actually don’t know what you did.

In addition, in EEG research is a common practice to normalize power values across the whole frequency range to avoid the effect of hemispherical asymmetries of power density of scalp rsEEG rhythms.

An interesting issue I just noticing in this second round is some incongruent information in the legend of the figures. So, in the text Authors stated that PSD was calculated from 1 to 500 ms after the stimulus onset. But, in Figure 8 the legend said that the averages were made form PSD values in the selected electrodes under resting-state!!!!! Moreover, in supplementary material in Figure 1 it states that the average is from the preparation state before go and stop stimuli. From Figures 2 to 10 the legends are again showing under resting state. I do not what this incongruence is due to, but this is not ok for the manuscript to be accepted.

Regarding comparisons and statistical analyses, the Authors have not answered my questions at all.

Finally, I still consider the conclusions are not sustained by data. Although as the Authors answer they have presented the classification results of each subject clearly in Tables 2-5, in the Discussion and Conclusions they based their ideas on the good accuracy results obtained in the classification of epochs in one or two subjects. This is obvious because the results for the rest are not so good. So the Authors are comparing the behavior of a classificatory system in one or two clean and well-recorded subjects when actually the rest of the results are not so good. Moreover, the results from ERP and ERSP are still not discussed, and the reasoning behind why they used PSD for classification instead of ERP is still missing, and this is surprising when they consider ERPs (i.e. N200 and P300) as the biomarkers of inhibition.

To sum up, I do not think the Authors had done substantial changes in the manuscript, I do not feel my questions and concerns had been answered and I do not consider this study suitable for publication. 

Author Response

We have revised the all statements of the manuscript carefully, according to reviewer suggestions. The authors are grateful to the editor and reviewer for the constructive comments on this manuscript. We believe all of the valuable suggestions have been addressed in our response and in the revised manuscript. 

Reviewer 2 Report

All queries have been successfully addressed.

Author Response

We would like to many thanks for reviewer suggestion and comments.